# Does Exercise Testing with Arm Crank Ergometer Substitute for Cycle Ergometer to Evaluate Exercise Capacity?

Miwako Deguchi [1,2], Hisayo Yokoyama [1,3,*], Nobuko Hongu [2,3], Atsuya Toya [1], Takahiro Matsutake [1,3], Yuta Suzuki [1,3], Daiki Imai [1,3], Yuko Yamazaki [4], Masanori Emoto [4] and Kazunobu Okazaki [1,3]

1 Department of Environmental Physiology for Exercise, Graduate School of Medicine, Osaka Metropolitan University, 3-3-138 Sugimoto, Sumiyoshi-ku, Osaka-shi 558-8585, Osaka, Japan; deguchi@omu.ac.jp (M.D.); d20ma007@st.osaka-cu.ac.jp (A.T.); matsutake@omu.ac.jp (T.M.); ysuzuki@omu.ac.jp (Y.S.); imai@omu.ac.jp (D.I.); kokazaki@omu.ac.jp (K.O.)
2 Department of Nutrition, Graduate School of Human Life and Ecology, Osaka Metropolitan University, 3-3-138 Sugimoto, Sumiyoshi-ku, Osaka-shi 558-8585, Osaka, Japan; kay.hongu@gmail.com
3 Research Center for Urban Health and Sports, Osaka Metropolitan University, 3-3-138 Sugimoto, Sumiyoshi-ku, Osaka-shi 558-8585, Osaka, Japan
4 Department of Metabolism, Endocrinology and Molecular Medicine, Graduate School of Medicine, Osaka Metropolitan University 1-4-3 Asahi-machi, Abeno-ku, Osaka-shi 545-8585, Osaka, Japan; z21717s@omu.ac.jp (Y.Y.); emoto-m@omu.ac.jp (M.E.)
* Correspondence: yokoyama_hisayo@omu.ac.jp; Tel.: +81-06-6605-2947

**Abstract:** Using the upper limbs to test cardiopulmonary exercise can be a useful option in the case of individuals who are unable to pedal a bicycle due to lower limb injury or disability. We evaluated whether exercise testing with the upper limbs can be used equivalently to that of the lower limbs in assessing exercise capacity. Nine collegiate rowers and eight collegiate cyclists underwent incremental exercise testing with an arm crank ergometer (ACE) and cycle ergometer (CE). Heart rate (HR) and oxygen uptake ($VO_2$) were monitored throughout the tests. Segmental muscle mass and flow-mediated dilation of brachial artery were measured to assess the training status of the upper limbs. The muscle mass of the brachium, upper limb, and trunk were greater in the rowers than in the cyclists ($p < 0.05$). The correlations between HR and $VO_2$ was significantly different depending on exercise modalities, ACE and CE, in both groups ($p < 0.001$). The estimated maximal $VO_2$ using the correlation formula and age-predicted maximal HR was significantly lower in the exercise testing group with ACE than in the group with CE in rowers and cyclists ($41.7 \pm 7.3$ vs. $52.6 \pm 8.6$ mL/kg/min, $p = 0.010$ and $35.5 \pm 14.2$ vs. $50.4 \pm 13.4$ mL/kg/min, $p = 0.011$, respectively). The results suggested that exercise capacity assessed by exercise testing with ACE is underestimated, regardless of the training status of the upper limbs. Further research is needed to verify factors which affect the correlations between HR and $VO_2$ during upper- and lower-limb exercise.

**Keywords:** exercise capacity; collegiate student athletes; training specificity; physiological responses; flow-mediate dilation; linear mixed effect regression model



## 1. Introduction

Assessments of cardiopulmonary function are critical in a variety of clinical settings. Before certain surgeries, especially major procedures, surgeons assess the patient's cardiopulmonary function in order to determine their ability to tolerate the physical stress of the surgery and the recovery process. Preoperative peak oxygen uptake (peak $VO_2$) is a significant predictor of postoperative mortality and the development of perioperative complications [1–3]. Some guidelines consider patients with peak $VO_2 < 10$ mL/kg/min as high risk and recommend minimally invasive surgery and nonsurgical treatment [4,5]. Furthermore, based on the result of cardiopulmonary exercise testing, physicians prescribe personalized exercise programs and quantitate the effect of therapeutic intervention of the patient in recovering from certain medical conditions [6,7].

Modes such as cycle ergometer (CE) and treadmill are widely used in such exercise testing [7,8]. However, these exercise modalities cannot be applied to individuals with lower extremity injuries or with disabilities in their lower limbs due to spinal cord injury and amputation. In such cases, exercise testing with an arm crank ergometer (ACE) can be a useful option. To date, several reports have shown that exercise testing with ACE has good reliability and validity in measuring aerobic capacity and is a suitable tool for assessing individual physical fitness and training effectiveness in healthy people [9] and in those with cardiovascular disease [10]. However, the method of applying the loading during exercise testing with ACE varies depending on the studies and no clear exercise test protocol with ACE has been fully established.

It is reported that maximal oxygen uptake ($VO_{2max}$) attained in exercise testing with ACE corresponded to about 60–70% of those attained in exercise testing with CE or treadmill walking [11–14]. However, we speculate that the subjects in their studies may simply not have been able to reach the exercise intensity that resulted in $VO_{2max}$ during exercise testing with ACE. Because heart rate (HR) shows a certain linear relationship with $VO_2$ during exercise testing over wide range of submaximal intensities [15], $VO_{2max}$ can be estimated as $VO_2$ corresponding to age-predicted maximal HR. Therefore, the estimated $VO_{2max}$ may be the same for exercise testing with ACE and CE if correlations between HR and $VO_2$ during exercise testing with ACE are equivalent for those with CE. If so, exercise testing with ACE can substitute for that with CE in evaluating exercise capacity. In that case, people with injuries or disabilities in their lower limbs will benefit from exercise testing with ACE. However, to date, it has not been elucidated whether the correlation between HR and $VO_2$ during exercise testing with ACE is equivalent to that during exercise testing with CE.

On the other hand, a few reports concluded that the difference between $VO_{2max}$ attained during upper- and lower-limb exercise was more modest in the individuals with well-trained upper limbs compared to those with untrained upper limbs [11,16]. Although these studies suggest the possibility that the training status of upper limbs might influence the correlations between HR and $VO_2$ during exercise testing with upper-limb exercise, there are no previous studies that examined whether that was true. Therefore, it needs to be clarified whether the presence of continuous, endurance activity of the upper limbs, such as in people who use manual wheelchairs, would affect the difference in the correlation between HR and $VO_2$ during exercise testing depending on exercise modalities. By doing so, useful suggestions may be provided for predicting $VO_{2max}$ obtained from lower-limb exercise tests from the results of exercise tests with the upper limbs.

The aim of this study is to elucidate whether the exercise testing with ACE can substitute for that of CE in the assessment of exercise capacity. To clarify this, this study included collegiate student athletes with or without trained upper limbs. We conducted an incremental exercise testing using ACE and CE in these subjects and examined whether the correlations between HR and $VO_2$ during exercise testing is affected by exercise modalities. We also tried to examine the effect of the training status of upper limbs on the correlation between HR and $VO_2$ during exercise testing with ACE. The novelty of this study is providing practical guidance of exercise testing with ACE and CE in the general population, including those with injuries and disabilities.

## 2. Materials and Methods

### 2.1. Subjects

Seventeen male collegiate student athletes (nine rowers and eight cyclists) were recruited from among the students in Osaka Metropolitan University (https://www.omu.ac.jp/en/ (accessed on 15 October 2023) [17]) by advertisement via e-mail. We considered the former as athletes with trained upper limbs, and the latter as athletes with trained lower limbs. The collegiate rowers were participating in any of the five events, single sculls, double sculls, pair, fours, and eight, depending on the tournament. All collegiate cyclists were road racers. The Quick Response (QR) code was used to register for the study. To calculate the sample size, we used G*Power software (version 3.1.9). The effect

size calculated from preliminary results (d = 1.09), the $\alpha$-error (0.05), and the power of $(1 - \beta = 0.80)$ were used for the detection of the difference estimated $VO_{2max}$ between exercise testing with ACE and CE in cyclist. Based on these assumptions, nine subjects were required for each group (rowers versus cyclists). This study was performed in accordance with the ethical guidelines of the 1975 Declaration of Helsinki. All subjects provided written informed consent before participating in the study, which was approved by the Institutional Review Board of the Osaka Metropolitan University Graduate School of Medicine (approval no.: 2022-128, approved on 10 November 2022). All trials for this study were also registered in the University Hospital Medical Information Network Clinical Trials Registry (UMIN 000048409).

### 2.2. Experimental Design

The study was conducted from December 2022 to April 2023. All the subjects visited the laboratory three times. They were instructed to finish meals at least two hours before visiting, and to avoid any strenuous physical activity and alcohol intake from the previous day. During the first visit, the subjects underwent anthropometric and flow-mediated dilation (FMD) measurements. They also conducted preliminary exercise testing with an arm crank ergometer (ACE), and filled an assessment sheet about daily training. Furthermore, we interviewed the participants about their smoking habits and family history of hypertension because these factors were thought to influence the result of FMD. In the second and third visit, the subjects underwent incremental exercise testing using ACE or a cycle ergometer (CE). The order of exercise testing was randomly determined using computer-generated random numbers (randomized crossover design), and these exercise testing methods were conducted at least five days apart to assure a full recovery for the participants.

### 2.3. Anthropometric Measurements

Height and weight were measured with a stadiometer and digital weighing scale, respectively. The body mass index (BMI) was calculated from height and weight using the following formula: BMI $[kg/m^2]$ = weight [kg]/(height [m])$^2$. Bioelectrical impedance analysis using a body composition analyzer (Physion MD; Nippon Shooter Ltd., Tokyo, Japan) estimated the percentage of body fat as well as segmental (brachium, forearm, thigh, lower leg and trunk) muscle mass. Bioelectrical impedance analysis estimates body composition based on the principle that adipose tissue has lower conductivity of electricity than muscle or bone [18]. The subjects were kept in the supine position. According to the guidance of the device, electrodes were attached to each of the following sites sequentially: wrist, radial point of elbow, ankle, and lateral cervical region of the knee on each side. In this way, measurements of body composition were conducted in each segment: trunk, brachium, forearm, thigh, and lower leg, as well as the whole body.

### 2.4. Assessment of Daily Training Status

Training status (type of exercise, training time per session, and number of sessions per week) was self-reported and was used for the calculation of training volume. Based on the compendium of physical activities [19], type of exercise and metabolic equivalents (METs) were confirmed. To give some examples: codes 02074 (MET value; 12.0, description; rowing, 200 watts) and code 01050 (MET value; 12.0, description; bicycling, 16–19 mph) were most frequently used to calculate training volume for rowers and cyclists, respectively. Training volume was determined using the following formula: METs-hour/week = training time per week [hour] × METs.

### 2.5. Flow-Mediated Dilation (FMD)

It was reported that athletes undergoing endurance or high-intensity resistance training showed greater flow-mediated dilation (FMD) of the brachial artery compared to age-matched sedentary controls [20,21]. Furthermore, Walther et al. reported that arm muscle mass and FMD of the brachial artery was greater in swimmers, with preferentially

trained upper limbs, than in cyclists with trained lower limbs [22]. Therefore, FMD data of the brachial artery were used as an indicator of the training status of upper limbs.

FMD value measurements were conducted in the brachial artery of the right brachium using a UNEXEF18G FMD device (Unex Corp., Nagoya, Japan) by a trained researcher. Room temperature was maintained at $22 \pm 2\,°C$ throughout the measurement. The subjects were instructed to assume the supine position. After the measurement of resting systolic blood pressure (SBP) and diastolic blood pressure (DBP) on the left upper arm, an inflation cuff was placed around their right forearm and two electrocardiogram clips were attached to both wrists. A probe was placed proximal to the elbow in a site that provided the clearest B-mode image of the anterior and posterior layers of intima. Then the cuff placed around the right forearm was inflated to 50 mmHg above resting SBP for five minutes. Diameter changes in the brachial artery were recorded during inflation and for two minutes after deflation while heart rate (HR) was recorded. The mean diameter of the brachial artery during the first 20 s after deflation was used as the baseline diameter. The maximal diameter during 120 s after deflation was also measured. FMD was automatically calculated from baseline and maximal post-deflation diameter using the following formula: FMD [%] = [(maximal post deflation diameter − baseline diameter)/baseline diameter] × 100.

### 2.6. Exercise Testing

The detailed protocols used for exercise testing with the arm crank ergometer (ACE) (881 E; Monark) and cycle ergometer (CE) (828 E; Monark) are shown below. After adjusting these ergometers to the subject, each exercise test was performed according to the protocol. Expiratory gas was continuously collected during the exercise testing using the electronic spirometry system integrated with a gas analyzer (AE-310S, Minato Medical Science, Osaka, Japan). This system measures oxygen consumption, carbon dioxide production, and respiratory exchange ratio (RER) on a breath-by-breath basis. The optical HR sensor was placed on the precordial region with a strap and it monitored HR throughout the exercise test (Polar H10, Polar Electro Oy, Kempele, Finland). Values at the end of each stage and one minute prior to the end of each stage were used to develop a regression equation for the correlation between HR and $VO_2$ for each exercise modality, ACE and CE. SBP and DBP were measured by the automated blood pressure monitor (EBP-330, Minato Medical Science, Osaka, Japan) throughout the test with CE. In the case of the exercise testing with ACE, SBP and DBP were measured only at the resting and recovery periods. Borg's scale was used every minute to evaluate the rating of perceived exertion (RPE, ranged from 6 to 20 points) [23]. Exercise was not continued any further when the participants met two of the following four criteria: (1) RPE exceeded 18, (2) HR exceeded 85% of the age-predicted maximum, (3) RER exceeded 1.1, and (4) the presence of a plateau in $VO_2$ in spite of the increase in loading or could not maintain the speed of rotation. After the termination, exercise without loading was continued for 3 min as a recovery period.

### 2.7. Arm Crank Ergometer (ACE)

When the subjects first visited the laboratory, they performed exercise with the arm crank ergometer (ACE) for familiarization and preliminary testing. The subjects were seated with their backs straight in a chair with a backrest and without casters and were instructed to keep their soles on the floor. The distance between the subjects and ACE was set up to ensure that their elbows were slightly bent when their hands were in the furthest position. Height of the ACE was adjusted for each subject and recorded so that the actual tests could be performed at the same height. Preliminary exercise testing was conducted to determine the loading protocol of ACE. After a rest for 5 min, the basic incremental protocol started with 12.5 watts (W) and subsequently increased by 12.5 W per stage [24]. The actual intensity from the second stage was determined for each subject to finish in the fourth stage. For instance, one subject was conducted at 12.5 W, followed by 25, 37.5, and 50 W, and another was conducted at 12.5 W, followed by 50, 62.5, and 75 W. Rotation

speed was set at 50 repetitions per minute (rpm), and exercise intensity was increased every three minutes.

### 2.8. Cycle Ergometer (CE)

We conducted exercise testing with a cycle ergometer (CE) according to the YMCA protocol after a rest for 5 min [25]. In short, all subjects pedaled at 25 W for the first three minutes. If the HR at the end of the first stage was less than 80, the loading in the second stage was set to 125 W. In the same way, the setting was 100 W for 80 to 90, 75 W for 90 to 100, and 50 W for over 100. Thereafter, the load was increased by 25 W until the fourth stage. Saddle height was set up to ensure that their knees were slightly bent when their foot was in the furthest position, and angle of the handle was adjusted for each subject. Rotation speed and exercise time per stage were set in the same manner as in the case of those matched with ACE.

### 2.9. Estimation of $VO_{2max}$ from the Exercise Testing with ACE and CE

As Figure 1 shows, $VO_{2max}$ was estimated from both the results of exercise testing with ACE and CE per subject, respectively. For each subject, regression equations representing the correlation between HR and $VO_2$ in each exercise modality were created. Using the regression equations, $VO_{2max}$ results were estimated as the $VO_2$ corresponding to the age-predicted maximal HR (HR = 220 − age).

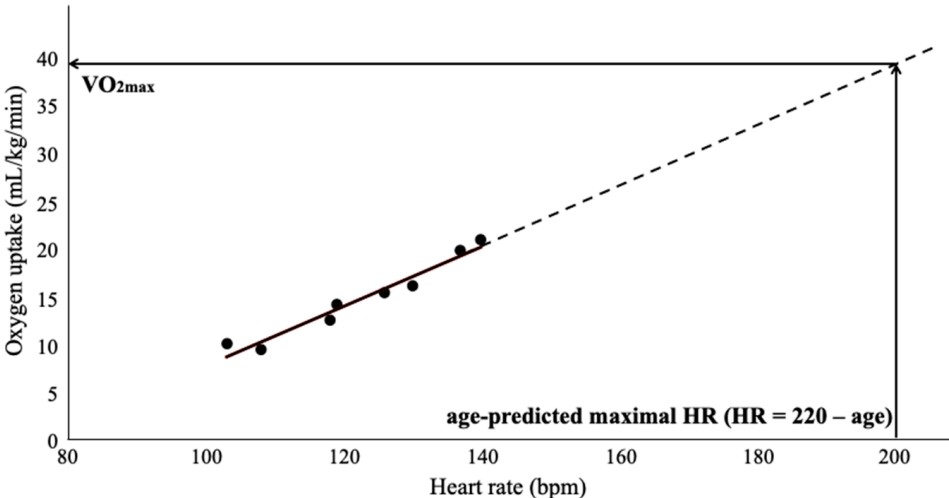

**Figure 1.** Estimation of $VO_{2max}$ from exercise testing with ACE and CE.

For each subject, values of HR and $VO_2$ at the end point and one minute prior to the end of the stage were used to create a regression equation in each exercise modality. Then, $VO_{2max}$ was estimated by substituting an age-predicted maximal HR (HR = 220 − age) into the obtained equation.

### 2.10. Statistics

We used an unpaired *t*-test to compare the parameters between the rowers and the cyclists. Spearman's rank correlation coefficient test was used to examine the correlation between HR and $VO_2$ during the exercise testing in each exercise modality, that is ACE and CE, in each group. The Linear Mixed Effect (LME) regression model was used in each athlete group to examine whether the correlation between HR and $VO_2$ during the exercise testing is modified by the exercise modalities. In the LME regression model, $VO_2$ was set as a dependent variable and HR, exercise modalities, and interaction between HR and exercise modalities as independent variables. A paired *t*-test was used to compare the estimated $VO_{2max}$ between the exercise modalities in the rowers and the cyclists. Linear regression analysis was carried out to detect the degree to which the estimated $VO_{2max}$ from exercise

testing with ACE contributed to that of exercise testing with CE. The estimated $VO_{2max}$ from exercise testing with CE was set as a dependent variable and that of exercise testing with ACE as an independent variable. All statistical analyses were performed with SPSS (version 27.0, IBM, New York, NY, USA). The significance level was set at *p* values less than 0.05.

## 3. Results

### 3.1. Characteristics of the Subjects

Table 1 shows the characteristics of the subjects. Age, height, weight, and BMI were similar in both groups. Percentage of body fat in the rowers was significantly lower than that in the cyclists. Muscle mass of brachium, upper limb (brachium + forearm), and trunk were greater in the rowers than in the cyclists. There were no statistically significant differences in the muscle mass of the forearm and lower extremities between the athlete groups. Resting systolic/diastolic blood pressures measured before exercise testing were not significantly different between the rowers and the cyclists. Neither years of the competition nor training volume were significantly different between the two groups. In both groups, most subjects (89% of rowers and 100% of cyclists) had been competing since the time of entering university. The rowers were preferentially performing resistance training such as push-ups and pull-ups and exercise with rowing ergometers as daily training, while the cyclists were conducting outdoor or indoor cycling on the roller stand as daily training. No subjects had a daily smoking habit. Each group had two subjects with a family history of hypertension.

**Table 1.** Characteristics of the subjects.

|  | Rowers (*n* = 9) | | | Cyclists (*n* = 8) | | | *p* |
|---|---|---|---|---|---|---|---|
| Age (years) | 19.4 | ± | 0.9 | 20.3 | ± | 1.0 | 0.104 |
| Height (cm) | 173.6 | ± | 7.4 | 174.5 | ± | 9.1 | 0.830 |
| Weight (kg) | 66.7 | ± | 7.2 | 66.2 | ± | 6.2 | 0.885 |
| Body fat (%) | 8.3 | ± | 3.1 | 12.4 | ± | 3.1 | 0.015 |
| BMI (kg/m$^2$) | 22.1 | ± | 1.5 | 21.8 | ± | 2.0 | 0.732 |
| Brachium $M_{musc}$ (kg/m$^2$) | 0.58 | ± | 0.10 | 0.46 | ± | 0.07 | 0.012 |
| Forearm $M_{musc}$ (kg/m$^2$) | 0.40 | ± | 0.04 | 0.36 | ± | 0.08 | 0.230 |
| UL $M_{musc}$ (kg/m$^2$) | 0.97 | ± | 0.14 | 0.82 | ± | 0.12 | 0.026 |
| Thigh $M_{musc}$ (kg/m$^2$) | 3.33 | ± | 0.57 | 3.06 | ± | 0.28 | 0.238 |
| Lower leg $M_{musc}$ (kg/m$^2$) | 1.02 | ± | 0.14 | 0.95 | ± | 0.11 | 0.271 |
| LL $M_{musc}$ (kg/m$^2$) | 4.36 | ± | 0.62 | 4.02 | ± | 0.30 | 0.184 |
| Trunk $M_{musc}$ (kg/m$^2$) | 5.19 | ± | 0.61 | 4.52 | ± | 0.59 | 0.035 |
| SBP (mmHg) | 111.9 | ± | 12.3 | 114.6 | ± | 14.0 | 0.673 |
| DBP (mmHg) | 73.2 | ± | 11.6 | 77.0 | ± | 12.6 | 0.528 |
| Years of competition (months) | 14.1 | ± | 9.9 | 18.8 | ± | 6.9 | 0.285 |
| Training volume (METs-hour/week) | 90.6 | ± | 86.2 | 92.2 | ± | 109.2 | 0.973 |

The data are presented as mean ± standard deviation (SD). These parameters were compared by an unpaired *t*-test. Abbreviations: BMI, body mass index; UL, upper limb (brachium + forearm); LL, lower limb (thigh + lower leg); $M_{mus}$, muscle mass; SBP, systolic blood pressure; DBP, diastolic blood pressure; METs, metabolic equivalents.

### 3.2. FMD of Brachial Artery

Table 2 shows the vascular structure and function of the brachial artery in rowers and cyclists. There were no statistically significant differences in baseline nor maximal deflation diameters between the rowers and the cyclists. FMD of brachial artery was not significantly different between the groups either.

**Table 2.** Vascular structure and function in rowers and cyclists.

|  | Rowers (*n* = 9) | | | Cyclists (*n* = 8) | | | *p* |
|---|---|---|---|---|---|---|---|
| $D_{base}$ (mm) | 4.3 | ± | 0.7 | 3.8 | ± | 0.3 | 0.151 |
| $D_{maxpostdefl}$ (mm) | 4.6 | ± | 0.7 | 4.1 | ± | 0.4 | 0.119 |
| FMD (%) | 7.9 | ± | 4.2 | 8.0 | ± | 2.7 | 0.976 |

The data are presented as mean ± SD. These parameters were compared by unpaired *t*-test. Abbreviations: $D_{base}$, baseline diameter; $D_{maxpostdefl}$, maximal post deflation diameter; FMD, flow-mediated dilation.

### 3.3. Correlation between HR and VO2 in Each Exercise Modality

Most cyclists (87.5%, *n* = 7) performed exercise testing according to the basic incremental protocol of ACE, starting at 12.5 W, followed by 25, 37.5, and 50 W. One cyclist performed exercise testing according to the modified protocol of ACE, which started with 12.5 W, followed by 37.5, 50, and 62.5 W. In the rowers, three subjects (33.3%) achieved 50 W as a maximal workload at the final (fourth) stage, three subjects (33.3%) achieved 62.5 W, two subjects (22.2%) achieved 72.5 W and one subject (11.1%) achieved 85 W. Next, we examined whether the correlations between HR and $VO_2$ during the exercise testing were modified by the exercise modalities, that is ACE and CE, in each group using the LME regression model. As shown in Figure 2, a significant interaction effect of HR × exercise modalities on $VO_2$ was found in both groups (both exhibited a significant difference at *p* < 0.001), suggesting that the correlations between HR and $VO_2$ during exercise testing were different depending on exercise modalities. Based on the LME regression model, the estimation formulae of $VO_2$ were as follows: $VO_{2\text{-ACE}} = 0.282 \times HR_{ACE} - 17.145$ (ρ = 0.814, *p* < 0.001) and $VO_{2\text{-CE}} = 0.361 \times HR_{CE} - 22.464$ (ρ = 0.811, *p* < 0.001) in rowers; $VO_{2\text{-ACE}} = 0.215 \times HR_{ACE} - 12.548$ (ρ = 0.702, *p* < 0.001) and $VO_{2\text{-CE}} = 0.313 \times HR_{CE} - 18.825$ (ρ = 0.594, *p* < 0.001) in cyclists.

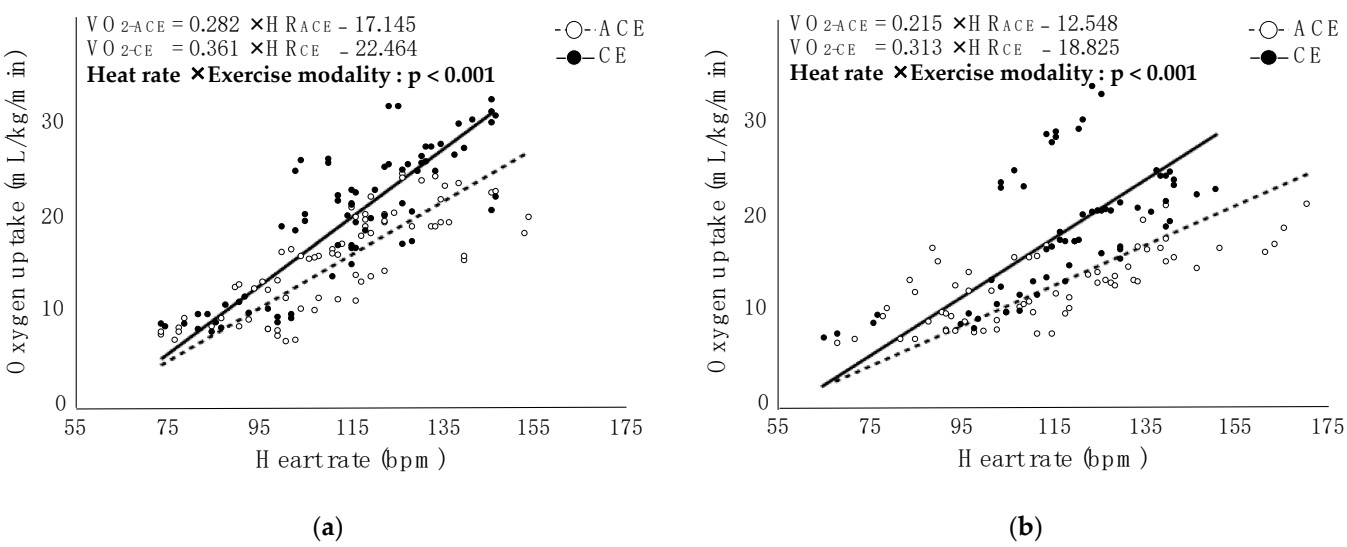

(**a**)  (**b**)

**Figure 2.** Correlation between HR and $VO_2$ in the exercise testing with ACE and CE in rowers (**a**) and cyclists (**b**). Values of HR and $VO_2$ at the end point and one minute prior to the end of the stage were plotted and used to create regression equations. The interaction effect between HR and exercise modalities on $VO_2$ was evaluated by the LME regression model. Abbreviations: ACE, arm crank ergometer; CE, cycle ergometer, $VO_2$, oxygen uptake; HR, heart rate.

### 3.4. Estimated VO2max from the Exercise Testing with ACE and CE

Table 3 shows the estimated $VO_{2max}$ from the results of the exercise testing with ACE and CE in rowers and cyclists. There were statistically significant differences between $VO_{2max}$ estimated from the results of exercise testing with each exercise modality in both groups. Based on the linear regression analysis in all subjects, the equations for the relation-

ship between estimated $VO_{2max}$ from each exercise test are as follows: $VO_{2max\text{-}CE} = 0.535 \times VO_{2max\text{-}ACE} + 30.826$ ($r^2 = 0.306$, $p = 0.021$).

**Table 3.** Estimated $VO_{2max}$ from the exercise testing with ACE and CE in rowers and cyclists.

| | Rowers (*n* = 9) | | | *p* | Cyclists (*n* = 8) | | | *p* |
|---|---|---|---|---|---|---|---|---|
| $VO_{2max\text{-}ACE}$ (mL/kg/min) | 41.7 | ± | 7.3 | 0.010 | 35.5 | ± | 14.2 | 0.011 |
| $VO_{2max\text{-}CE}$ (mL/kg/min) | 52.6 | ± | 8.6 | | 50.4 | ± | 13.4 | |

The data are presented as mean ± SD. Estimated $VO_{2max}$ from the exercise testing with ACE and CE in rowers and cyclists were compared by paired *t*-test. Abbreviations: ACE, arm crank ergometer; CE, cycle ergometer; $VO_{2max}$, maximal oxygen uptake.

## 4. Discussion

The purpose of this study was to elucidate whether exercise testing with ACE can be substituted for that of CE in the assessment of exercise capacity. Therefore, we tried to clarify whether the correlations between HR and $VO_2$ during exercise testing were different depending on the exercise modalities, i.e., ACE and CE. Our main findings were that the correlations between HR and $VO_2$ during the submaximal exercise testing were different depending on exercise modalities both in collegiate rowers and collegiate cyclists and that their $VO_{2max}$ estimated from exercise testing with ACE was significantly lower than that with CE.

As we have mentioned, it is sometimes difficult for people with disabilities or with orthopedic problems in their lower limbs to perform exercise testing with the lower extremities. For example, in a study which examined the feasibility of exercise testing in patients prior to total hip or knee arthroplasty, approximately 40% of them could not perform exercise testing with CE [26]. Therefore, if the exercise testing with ACE can substitute for that of CE, people with disabilities or with orthopedic problems in their lower limbs will benefit from exercise testing with ACE in assessing preoperative cardiopulmonary function. Exercise testing with ACE may also be useful for this population in evaluating the cardiopulmonary fitness for the purpose of understanding postoperative rehabilitation effects. The above was the reason it was necessary to verify whether exercise testing with ACE can be a substitute for exercise testing with CE.

So far, some studies tried to compare the physiological responses during exercise testing between ACE and CE or treadmill walking and reported that peak $VO_2$ obtained by exercise testing with ACE was lower than that with CE or treadmill [12–14]. However, the subjects in these studies may simply have not reached the maximum cardiopulmonary loading during the exercise test with ACE due to muscle fatigue of the upper limbs [13]. In addition, HR at the end of the exercise testing with ACE was lower than that obtained with CE or treadmill, while RPE was similar or higher [12,13]. Therefore, if the peak $VO_2$ at the maximum cardiopulmonary load can be predicted by exercise testing with ACE, exercise testing with ACE can be useful option, instead of that of CE, to evaluate cardiopulmonary fitness. From the above background, we tried to determine the correlation between HR and $VO_2$ during exercise testing for each exercise modality, ACE and CE, and estimated the $VO_{2max}$ as the $VO_2$ corresponding to age-predicted maximal HR using the correlation formula. Contrary to our expectations, however, the correlations between HR and $VO_2$ were different depending on the exercise modalities, and $VO_2$ corresponding to the same HR was constantly smaller during the exercise testing with ACE than that during the exercise testing with CE. These results suggest that $VO_{2max}$ may be underestimated in the case of using exercise testing with ACE compared to that of CE.

Cardiopulmonary exercise testing is used in various clinical settings. For instance, such exercise testing is used as one of the measures for detecting coronary artery disease. In this case, exercise testing with ACE will be a useful option for people with injuries or disabilities in their lower limbs because it is enough to meet the purpose as long as the subject can reach an exercise load that causes any symptoms or abnormal findings to be displayed on an electrocardiogram. Likewise, exercise testing with ACE will benefit

them in the context of evaluating the effects of cardiopulmonary training or rehabilitation. This is because if the $VO_2$ corresponding to the same HR increases, it can be confirmed that such training or rehabilitation was effective. On the other hand, taking our results into consideration, some cautions will be required when using exercise testing with ACE for preoperative risk assessment. If a patient is judged as inoperative because of an underestimation of cardiopulmonary function by exercise testing with ACE, it leads to losing an opportunity to receive appropriate treatment. It may also result in unnecessary occupancy of intensive care unit after surgery. To date, several regression equations have been developed to estimate peak $VO_2$ during lower-limb exercise based on peak $VO_2$ obtained from upper-limb exercise [12,13,27]. Although these regression equations could explain about 80% of the peak $VO_2$ obtained from lower-limb exercise by using not only peak $VO_2$ obtained from upper-limb exercise but also gender, body weight, or lean body mass as independent variables, such regression equations were developed mainly for healthy subjects and have not been fully validated. Likewise, the present study targeted healthy, collegiate athletes. However, the results from the linear regression analysis showed that $VO_{2max\text{-}ACE}$ could explain only about 30% of the $VO_{2max\text{-}CE}$. When using exercise testing with ACE as a preoperative test in the future, especially in major, invasive surgeries, it will be necessary to explore a specific cutoff value for peak $VO_2$ or estimated $VO_{2max}$ to determine surgical indications. To this end, it is needed to explore factors affecting the relationship between $VO_{2max}$ estimated from exercise testing with ACE and that of CE with a more diverse population.

The present study was unable to clarify the physiological mechanisms which were responsible for the differences in the correlations between HR and $VO_2$ depending on exercise modalities. Previous studies reported differences in hemodynamic response during exercise between upper and lower limbs. The higher arterial pressure and the lower stroke volume were observed during upper-limb exercise compared to lower-limb exercise [28,29]. Moreover, HR needed to meet the same oxygen demand was higher during the exercise testing with ACE than during the exercise testing with CE [30]. These results suggest that during upper-limb exercise, higher afterload due to greater arterial pressures leads to lower stroke volume, which results in the increased HR to compensate for the decreased stroke volume. In addition, the greater sympathetic stimulation during upper-limb exercise may be related to the elevated HR [31]. The increased HR during upper-limb exercise may be responsible for the differences in the correlations between HR and $VO_2$ depending on the exercise modalities found in the present study.

In the present study, we examined the effect of the training status of upper limbs on the correlation between HR and $VO_2$ during exercise testing with ACE. Based on segmental muscle mass and maximal workload achieved during exercise testing with ACE, we considered collegiate rowers as the individuals with trained upper limbs. Our results showed that for both collegiate rowers and collegiate cyclists, the correlation between HR and $VO_2$ during exercise testing for each exercise modality was different and that estimating the $VO_{2max}$ as the $VO_2$ corresponding to age-predicted maximal HR using the correlation formula was lower in the exercise testing with ACE than in that of CE. These results suggested that exercise capacity assessed by exercise testing with ACE is underestimated, regardless of the training status of the upper limbs. Previous studies reported that the difference in $VO_{2max}$ obtained between exercise testing with ACE and CE was more modest in individuals with well-trained upper limbs [11,16]. These studies compared $VO_{2max}$ at the end of exercise testing for each exercise modality. Therefore, it may be inappropriate to directly compare the results of our study that examined the correlation between HR and $VO_2$ during exercise testing with theirs; however, our study suggests that exercise capacity is underestimated in the case of using results obtained from exercise testing with ACE, even in individuals with trained upper limbs. In future studies, to determine the extent to which $VO_{2max}$ is underestimated in the exercise testing with ACE in various populations may lead to the development of a prediction equation for $VO_{2max}$ obtained from exercise testing with CE based on the results from exercise testing with ACE.

A few limitations of the present study should be noted. First, it concerns the background of the subjects. The subjects had a short history of competition. Six of the nine rowers (66.7%) had been competing for less than a year. In the cyclists, one subject (12.5%) had been competing for less than one year and five (62.5%) had been competing for one to two years. The subjects recruited in the present study are considered to be in the process of developing exercise training adaptation. Further studies should be conducted with more trained athletes. Additionally, the percentage of body fat in the rowers was significantly lower than that in the cyclists. The difference in body composition could have had a significant impact on the main research findings. The second point concerns how to obtain $VO_{2max}$ in the present study. In order to achieve our study aim, it was needed to examine whether the correlations between the HR and $VO_2$ formula during exercise testing were equivalent between ACE and CE. Therefore, $VO_{2max}$, which we used in the present study, were estimated values obtained from the formula, not actual measurement data. On the other hand, actual measurement values of $VO_{2max}$ reflect more biological, physiological information than the estimated values. In fact, previous studies have reported that peak $VO_2$ during lower-limb exercise can be predicted with high accuracy using peak $VO_2$ obtained from upper-limb exercise and other parameters. Using actual measured $VO_{2max}$ during exercise testing with ACE could have been more useful in predicting $VO_{2max}$ during exercise testing with CE. Third, this study did not evaluate hemodynamic response and sympathetic nerve activity during exercise testing. Future studies are needed to clarify whether these effect the differences in the correlations between HR and $VO_2$ depending on the exercise modalities. Finally, FMD data of the brachial artery measured as an indicator of the training status of upper limb were not significantly different between rowers and cyclists. Exercise training induced arterial remodeling such as decreased wall thickness and dilation of vessel diameter to adapt to increased blood pressure and blood flow during exercise [32,33]. Some studies suggest that this remodeling may be responsible for the lower FMD response in athletes [34,35]. In the present study, however, there were no statistically significant differences in wall thickness and baseline diameter of brachial artery between the two groups ($p$ = 0.754 and 0.151, respectively). This indicates that the FMD of the brachial artery does not reflect the training status of the upper limbs as assessed by segmental muscle mass and maximal workload achieved during exercise testing with ACE. Further research should use another indicator to assess the training status of the upper limbs.

## 5. Conclusions

The correlations between HR and $VO_2$ during the submaximal exercise testing was different depending on exercise modalities, ACE and CE, both in collegiate rowers and collegiate cyclists, and their estimated $VO_{2max}$ as the $VO_2$ corresponding to age-predicted maximal HR using the correlation formula was lower in exercise testing with ACE than with CE in both groups. These results suggest that exercise testing with ACE underestimates exercise capacity regardless of the training status of the upper limbs. While exercise testing with ACE will benefit in some situations, such as the evaluation of the effect of cardiopulmonary training and detection of coronary artery disease, our results suggest that the exercise testing with ACE cannot substitute that of CE, at least for the assessment of $VO_{2max}$. Furthermore, it is difficult to predict $VO_{2max}$ obtained from exercise testing with CE based on the results from exercise testing with ACE and the condition of the upper limbs. Practitioners involved in exercise testing need to keep this in mind. Further research is needed to investigate what factors influence the difference in the correlation between HR and $VO_2$ during exercise testing for each exercise modality in order to apply the exercise testing with ACE instead of CE to assess exercise capacity.

**Author Contributions:** Conceptualization, M.D. and H.Y.; methodology, M.D. and H.Y.; validation, M.D. and H.Y.; formal analysis, M.D. and H.Y.; investigation, M.D., H.Y., A.T. and Y.Y.; resources, Y.Y and M.E.; data curation, M.D., H.Y., T.M., Y.S., D.I. and K.O.; writing—original draft preparation, M.D., H.Y. and N.H.; writing—review and editing, H.Y., N.H., A.T., T.M., Y.S., D.I., Y.Y., M.E. and

K.O.; visualization, M.D. and H.Y.; supervision, H.Y. and N.H.; project administration, H.Y.; funding acquisition, M.D. All authors have read and agreed to the published version of the manuscript.

**Funding:** This research was funded in part by FY2022 Collaborative Research Grant for Cooperative Research from Osaka Metropolitan University (No. 2022-4).

**Institutional Review Board Statement:** The study was conducted in accordance with the Declaration of Helsinki, and approved by the Institutional Review Board of the Osaka Metropolitan University Graduate School of Medicine (Approval no.: 2022-128, approved on 10 November 2022).

**Informed Consent Statement:** Written informed consent was obtained all the subjects.

**Data Availability Statement:** The data presented in this study are openly available in FigShare at https://doi.org/10.6084/m9.figshare.24441205 (accessed on 1 December 2023), reference number [36].

**Acknowledgments:** We thank Munehiro Tetsuguchi (Osaka Kyoiku University) for helpful advice. We acknowledge our participants for their time and cooperation in this investigation.

**Conflicts of Interest:** The authors declare no conflict of interest.

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
