# Peer review of "Does Exercise Testing with Arm Crank Ergometer Substitute for Cycle Ergometer to Evaluate Exercise Capacity?"

_applsci, doi:10.3390/app132312926_

Round 1

Reviewer 1 Report

Comments and Suggestions for Authors

The importance of providing a family history of hypertension in approximately 25% of the subjects should be explained. While the body weights, heights and Body Mass Indexes of cyclists and rowers are similar, the difference in fat percentages seems high (12% versus 1%). I couldn't understand this difference. It is almost a known fact that ACE exercise testing cannot replace exercise testing with CE. By performing a regression analysis between the two tests, it could be determined by what percentage one would affect the other. Again, it is an expected result that the exercise test performed with ACE may underestimate exercise capacity. I thought there might be a suggestion to add a fixed number to the result so that the test can show the real level.

Author Response

We appreciate Reviewer 1 for the kind and important comments. Our point-by-point responses are as below.

The importance of providing a family history of hypertension in approximately 25% of the subjects should be explained.

In this study, we interviewed about smoking habit and family history of hypertension because these factors were thought to influence the result of flow-mediated dilation (FMD). Prevalence of family history of hypertension was not different between the two groups.

According to Reviewer 1’s comments, we have revised the corresponding sentence as follows:

Furthermore, we interviewed about smoking habit and family history of hypertension because these factors were thought to influence the result of FMD.  (Lines 120-122)

 Each group had two subjects with a family history of hypertension (Line 263)

While the body weights, heights and Body Mass Indexes of cyclists and rowers are similar, the difference in fat percentages seems high (12% versus 1%). I couldn't understand this difference.

As we showed in the original manuscript, the percentage of body fat was 8.3 ± 3.1 % in rowers, and 12.4 ± 3.1 % in cyclists, respectively. However, since the difference of body composition could have modulated our research findings, we added the above content as limitation to the revised manuscript.

(Lines 414-422)

It is almost a known fact that ACE exercise testing cannot replace exercise testing with CE. By performing a regression analysis between the two tests, it could be determined by what percentage one would affect the other. Again, it is an expected result that the exercise test performed with ACE may underestimate exercise capacity. I thought there might be a suggestion to add a fixed number to the result so that the test can show the real level.

Thank you again for Reviewer 1’s meaningful suggestion. According to the suggestion, we conducted linear regression analysis. In addition, we discussed the results obtained.

We added the above content to the revised manuscript as follows:

Linear regression analysis was carried out to detect how much degree the estimated VO2max from exercise testing with ACE contributed to that from exercise testing with CE. Estimated VO2max from exercise testing with CE and ACE were set as a dependent variable and an independent variable, respectively. (Lines 237-241)

Based on the linear regression analysis in all subjects, equations for the relationship between estimated VO2max from each exercise testing was as follows: VO2max-CE = 0.535 × VO2max-ACE + 30.826 (r2 = 0.306, p = 0.021). (Lines 309-311)

 Although these regression equations could explain about 80 % of peak VO2 obtained from lower limb exercise by using not only peak VO2 obtained from upper limb exercise but also gender, body weight, or lean body mass as independent variables, such regression equations were developed mainly for healthy subjects and have not been fully validated. Likewise, the present study targeted healthy, collegiate athletes. However, the results from the linear regression analysis showed that VO2max-ACE could explain only about 30 % of the VO2max-CE. When using exercise testing with ACE as a preoperative test in the future, especially in major, invasive surgeries, it will be necessary to explore a specific cutoff value for peak VO2 or estimated VO2max to determine surgical indications. To this end, it is needed to explore factors affecting the relationship between VO2max estimated from exercise testing with ACE and that with CE with more diverse population.

(Lines 369-380)

Reviewer 2 Report

Comments and Suggestions for Authors

In the present study, the authors have posed a noteworthy research question, and the study design appears to be generally appropriate. However, there are several crucial limitations that warrant more consistent attention.

Firstly, the novelty and aim of the study are not clearly delineated. The introduction transitions from describing clinical to sport-related conditions, creating ambiguity. Clarifying the focus and novelty early on would enhance the overall coherence of the study.

A significant limitation lies in the methodological approach, particularly the estimation, rather than measurement, of VO2max. The authors should consider addressing this limitation by exploring alternative or complementary methods to enhance the accuracy of their findings.

Furthermore, a notable concern arises in the discussion regarding the target audience for these results. It remains unclear whether the implications are directed solely towards athletes or extend to other populations. Considering that the data primarily derive from athletes, it may be inappropriate to generalize findings to individuals with disabilities or clinical conditions. Therefore, the authors should establish early on, starting from the introduction, whether the focus is exclusively on performance-related considerations for athletes or if there is an intention to broaden the scope to include other populations. In the latter case, a comprehensive explanation of why and how these data can be extrapolated to diverse populations should be provided.

In summary, while the study holds promise, addressing these limitations and enhancing the clarity and consistency in the framing of the research question, novelty, and target audience will significantly strengthen the overall impact and contribution of the manuscript.

Comments on the Quality of English Language

Authors may consider to double check for typos as I spotted a few.

Author Response

We appreciate Reviewer 2 for the kind and important comments. Our point-by-point responses are as below.

Firstly, the novelty and aim of the study are not clearly delineated. The introduction transitions from describing clinical to sport-related conditions, creating ambiguity. Clarifying the focus and novelty early on would enhance the overall coherence of the study.

As Reviewer 2 pointed out, we needed to refer to our focus and novelty more clearly.  In order to examine whether the activity level of upper limb affects the possibility of ACE to substitute for CE, we  studied collegiate athletes with different upper limb training conditions. We aim to expand our results to the general population, including those with injuries and disabilities. Therefore, we have re-structured the Introduction section to clarify them and revised the sentence of the beginning of the Discussion section as follows:

The purpose of this study was to elucidate whether the exercise testing with ACE can substitute for that with CE in the assessment of exercise capacity. Therefore, we tried to clarify whether the correlations between HR and VO2 during exercise testing were different depending on the exercise modalities, i.e., ACE and CE. (Lines 318-321)

A significant limitation lies in the methodological approach, particularly the estimation, rather than measurement, of VO2max. The authors should consider addressing this limitation by exploring alternative or complementary methods to enhance the accuracy of their findings.

We really appreciate Reviewer 2’s suggestion. As we mentioned in the Introduction section, our aim was to examine whether the exercise testing with ACE can substitute for CE and  evaluate their exercise capacity. In order to achieve this aim, it was needed to examine whether the correlations formula between HR and VO2 during exercise testing were equivalent between ACE and CE. We speculate that the subjects may have not been able to exercise to their maximum cardiopulmonary loading during exercise testing especially with ACE. Therefore, we intentionally did not measure VO2max in the exercise testing.

We have re-structured the Introduction section to show the above-mentioned contents as our strength. We also added a sentence to the Discussion section as follows:

In the future study, to determine the extent to which VO2max is underestimated in the exercise testing with ACE in various population may lead to the development of prediction equation for VO2max obtained from exercise testing with CE based on the results from exercise testing with ACE. (Lines 410-413)

On the other hand, actual measurement value of VO2max could be more useful in some context; actual measurement value of VO2max reflect more biological, physiological information than estimated value. For example, previous studies have reported that peak VO2 during lower limb exercise can be predicted with high accuracy using peak VO2 obtained from upper limb exercise and other parameters. We added the above content as limitation to the revised manuscript as follows:

The second point concerns how to obtain VO2max in the present study. In order to achieve our study aim, it was needed to examine whether the correlations between HR and VO2 formula during exercise testing were equivalent between ACE and CE. Therefore, VO2max we used in the present study were estimated values obtained from the formula, not actual measurement value. On the other hand, actual measurement value of VO2max reflects more biological, physiological information than estimated value. In fact, previous studies have reported that peak VO2 during lower limb exercise can be predicted with high accuracy using peak VO2 obtained from upper limb exercise and other parameters. Using actual measured VO2max during exercise testing with ACE could have been more useful in predicting VO2max during exercise testing with CE.  (Lines 422-431)

Furthermore, a notable concern arises in the discussion regarding the target audience for these results. It remains unclear whether the implications are directed solely towards athletes or extend to other populations. Considering that the data primarily derive from athletes, it may be inappropriate to generalize findings to individuals with disabilities or clinical conditions. Therefore, the authors should establish early on, starting from the introduction, whether the focus is exclusively on performance-related considerations for athletes or if there is an intention to broaden the scope to include other populations. In the latter case, a comprehensive explanation of why and how these data can be extrapolated to diverse populations should be provided.

We really appreciate the important suggestion by Reviewer 2. According to  Reviewer 2’s comments, we have re-structured the Introduction section to refer clearly that this study has the potential to provide practical guidance of exercise testing with ACE and CE in general population, including those with injuries and disabilities.

Authors may consider to double check for typos as I spotted a few.

We reviewed throughout the text again so as to confirm our English appropriate.

Reviewer 3 Report

Comments and Suggestions for Authors

Thank you for the opportunity to review a topic, namely, Does Exercise Testing with Arm Crank Ergometer Substitute for Cycle Ergometer to Evaluate Exercise Capacity?.

This paper is interesting.

However, I have some concerns, as follows:

1. Introduction: add the novelty of this study.

2. Line 85: "Seventeen male collegiate student athletes (9 rowers and 8 cyclists) were recruited from the members who regularly participated in extracurricular sports".

It seems necessary to detail the exact disciplines, which rowing was divided into.

Furthermore, the Authors should add description of cycling performance, as follows: BMX; Tandem Cycling; Cyclocross; Track;· Mountain; or Road cycling.

3. Line 119: "Bioelectrical impedance analysis using body composition analyzer (Physion MD; Nippon Shooter Ltd., Tokyo, Japan) ".

The authors is asked to detail this assessment method (on what basis was the assessment based?), procedures and equations that have been applied to determine the body fat percentage. Is it really possible to measure muscle mass using the BIA method?

4. Line 125: "based on the Compendium of physical activities [19], type of exercise and metabolic equivalents (METs) were confirmed".

It seems necessary to specify codes for specific sporting activities and specific MET values used by authors in this study.

5. Line 221: "Statistics

I suggest the authors must add regression model fitting criteria, too.

Additionally, there are three forms of Student's t-test. Hence, which one was specifically used in this study?

6. Line 263 “Next, we examined whether the correlations between HR and VO2 during the exercise testing is modified by the exercise modalities“.

The section “Statistics” does not cover any statistical methods for the assessment of correlation. This information is requested to be supplemented as well as explained how the correlation between the variables was assessed.

7. Lines 240-244: Table 1 "Percentage of body fat in the rowers was significantly lower than that in the cyclists“ (8.3% vs. 12.4% ).

How can authors explain such surprising and controversial findings of this study?

8. Furthermore, If the authors performed an experimental study, all subjects had to be homogeneous before randomization. In the present study, study participants were different by body composition components. This limitation, as a confounder, could have had a significant impact on the main research findings. Thus, the authors must explain all this in the manuscript.

9. The conclusions lack concrete practical guidance on how to apply the study results were obtained.

Kind regards

Author Response

We appreciate Reviewer 3 for the kind and important comments. Our point-by-point responses are as below.

  1. Introduction: add the novelty of this study.

In accordance with Reviewer 3’s suggestion, we have re-structured Introduction section to clarify the novelty of this study.

  1. Line 85: "Seventeen male collegiate student athletes (9 rowers and 8 cyclists) were recruited from the members who regularly participated in extracurricular sports".

It seems necessary to detail the exact disciplines, which rowing was divided into.

Furthermore, the Authors should add description of cycling performance, as follows: BMX; Tandem Cycling; Cyclocross; Track;· Mountain; or Road cycling.

According to Reviewer 3’s comments, we added the descriptions about the categories of each athletic events in which our collegiate athletes were participating as follows:

The collegiate rowers were participating in any of the five events, single sculls, double sculls, pair, fours, and eight, depending on the tournament. All collegiate cyclists were road racers. (Lines 98-100)

  1. Line 119: "Bioelectrical impedance analysis using body composition analyzer (Physion MD; Nippon Shooter Ltd., Tokyo, Japan) ".

The authors is asked to detail this assessment method (on what basis was the assessment based?), procedures and equations that have been applied to determine the body fat percentage. Is it really possible to measure muscle mass using the BIA method?

According to Reviewer 3’s comments, we added the descriptions about the principle of BIA and measurement procedures using Physion MD as follows:

Bioelectrical impedance analysis estimates body composition based on the principle that adipose tissue is significantly less conductive than muscle or bone [18]. The subjects were kept in the supine position. According to the guidance of the device, electrodes were attached to each of the following sites sequentially: wrist, radial point of elbow, ankle, and lateral cervical region of the knee on each side. In this way, measurements of body composition were conducted in each segment: trunk, brachium, forearm, thigh, and lower leg, as well as whole body. (Lines 133-139)

According to the above revisions, we newly added reference #18.

  1. Line 125: "based on the Compendium of physical activities [19], type of exercise and metabolic equivalents (METs) were confirmed".

It seems necessary to specify codes for specific sporting activities and specific MET values used by authors in this study.

According to Reviewer 3’s suggestion, we added the code, MET value, and description most frequently used to calculate training volume in rowers and cyclists, respectively as follows:

To give some examples: codes 02074 (MET value; 12.0, Description; Rowing, 200 Watt) and code 01050 (MET value; 12.0, Description; Bicycling, 16-19 mph) were most frequently used to calculate training volume for rowers and cyclists, respectively. (Lines 145-149)

  1. Line 221: "Statistics“

I suggest the authors must add regression model fitting criteria, too.

Additionally, there are three forms of Student's t-test. Hence, which one was specifically used in this study?

  1. Line 263 “Next, we examined whether the correlations between HR and VO2 during the exercise testing is modified by the exercise modalities“.

The section “Statistics” does not cover any statistical methods for the assessment of correlation. This information is requested to be supplemented as well as explained how the correlation between the variables was assessed.

Regarding Reviewer 3’s comments, the parameters between the rowers and the cyclists were compared by unpaired t-test. The correlations between HR and VO2 during the exercise testing were examined by Spearman’s rank correlation coefficient test.

We have revised and added the above content to the revised manuscript as follows:

We used unpaired t-test to compare the parameters between the rowers and the cyclists. Spearman’s rank correlation coefficient test examined the correlation between HR and VO2 during the exercise testing in each exercise modality, that is ACE and CE, in each group. (Lines 234-237)

These parameters were compared by unpaired t-test. (Lines 267-268)

These parameters were compared by unpaired t-test. (Line 278)

Based on the LME regression model, the estimation formula of VO2 were as follows: VO2-ACE = 0.282 × HRACE 17.145 (r = 0.814, p < 0.001) and VO2-CE = 0.361 × HRCE 22.464 (r = 0.811, p < 0.001) in rowers; VO2-ACE = 0.215 × HRACE 12.548 (r = 0.702, p < 0.001) and VO2-CE = 0.313 × HRCE 18.825 (r = 0.594, p < 0.001) in cyclists. (Lines 294-298)

  1. Lines 240-244: Table 1 "Percentage of body fat in the rowers was significantly lower than that in the cyclists“ (8.3% vs. 12.4% ).

How can authors explain such surprising and controversial findings of this study?

  1. Furthermore, If the authors performed an experimental study, all subjects had to be homogeneous before randomization. In the present study, study participants were different by body composition components. This limitation, as a confounder, could have had a significant impact on the main research findings. Thus, the authors must explain all this in the manuscript.

In the present study, we recruited the subjects with trained upper limb (rowers) and without trained upper limb (cyclist) to each study group in order to achieve our research aim mentioned above. As a result, there was a significant difference in the percentage of body fat as well as muscle mass of upper extremity between the two groups. The differences in body composition between the groups might’ve been derived from their daily training methods: the rowers were preferentially performing resistance training such as push-ups and pull-ups and exercise with rowing ergometer as daily training, while the cyclists were conducting outdoor cycling or indoor one on the roller stand as daily training. As Reviewer 3 pointed out, this difference in body composition between the groups may have influenced the results of this study. We have added the content mentioned above to the limitation paragraph. (Lines 414-422)

  1. The conclusions lack concrete practical guidance on how to apply the study results were obtained.

Our main findings were that the correlations between HR and VO2 during the submaximal exercise testing was different depending on exercise modalities and that VO2max estimated from exercise testing with ACE was significantly lower than that with CE regardless of training status of upper limb. These results suggest that the exercise testing with ACE cannot substitute for that with CE, at least for the assessment of VO2max. Furthermore, it is difficult to predict VO2max obtained from exercise testing with CE based on the results from exercise testing with ACE and activity level of upper limb. On the other hand, exercise testing with ACE will benefit in some situations such as evaluation of the effect of cardiopulmonary training and detection of coronary artery disease. We added the above content to the Conclusions as practical guidance to the revised manuscript as follows:

While exercise testing with ACE will benefit in some situations such as evaluation of the effect of cardiopulmonary training and detection of coronary artery disease, our results suggest that the exercise testing with ACE cannot substitute for that with CE, at least for the assessment of VO2max. Furthermore, it is difficult to predict VO2max obtained from exercise testing with CE based on the results from exercise testing with ACE and condition of upper limb. Practitioners involved in exercise testing need to keep this in mind.  (Lines 452-458)

Round 2

Reviewer 3 Report

Comments and Suggestions for Authors

The Authors answered my questions and improved the manuscript perfectly.

All in all, the paper is interesting.

However, I have one concern:

Line 275: If Authors used Spearman’s rank correlation coefficient, then “r“ must be changed to “rho“ (please see information on lines 338-340).

Best regards

Author Response

Thank you for Reviewer 3’s kind review again.

Line 275: If Authors used Spearman’s rank correlation coefficient, then “r“ must be changed to “rho“ (please see information on lines 338-340).

As Reviewer 3 poited out, we had to use Spearman’s rank correlation coefficient “rho”, not “r”.

We re-revised the manuscript as follows:

Line 295

Based on the LME regression model, the estimation formula of VO2 were as follows: VO2-ACE = 0.282 × HRACE − 17.145  (ρ = 0.814, p < 0.001) and VO2-CE = 0.361 × HRCE − 22.464 (ρ = 0.811, p < 0.001) in rowers; VO2-ACE = 0.215 × HRACE − 12.548 (ρ = 0.702, p < 0.001) and VO2-CE = 0.313 × HRCE − 18.825 (ρ = 0.594, p < 0.001) in cyclists.